# Proof Search Augmented Language Models

## Abstract

Transformer language models (TLMs) exhibit an impressively general range of capabilities. A growing body of work aims to harness these models for complex reasoning problems expressed in natural language. However, recent theoretical and empirical results have revealed limits to the algorithmic generalization of TLM reasoning. Transformers trained to solve deduction problems from one distribution fail to solve instances of the same problem type drawn from other distributions. We propose to improve the systematic reasoning capabilities of TLMs via a differentiable proof search module, yielding proof-search augmented language models (PSALMs). In a PSALM, a Transformer is responsible for predicting rule and fact representations for a neural theorem prover (NTP). The NTP performs a backward-chaining search over proofs, scoring them based on a soft unification operation. Our results show that PSALMs successfully generalize in deduction tasks where vanilla transformers do not learn systematic behavior, can be adapted to more natural text with only label supervision, and robustly handle large examples where proprietary LLMs make mistakes.

## 1 Introduction

The general utility of large language models for text- and code-based tasks is a major factor driving their increasing adoption. Pursuant to this, there is a growing premium placed on their ability to 'reason' in order to widen the range of tasks they can handle. Reasoning in this context translates to following rules, integrating information in a consistent way, and being able to solve complex problems. One big challenge is search: strategies like chain-of-thought inference (Wei et al., 2022), in which models generate intermediate steps to break problems down, are fundamentally greedy and can leave models in dead-ends after they commit to an inconsistent rationale. Strategies like tree-of-thought (Yao et al., 2023) that allow backtracking have to navigate the search space of token strings, which is massive, and still fundamentally depend on the model to propose consistent steps.

This work aims to bridge the gap between classical proof search in systems like Prolog and the soft reasoning capabilities of transformers. Such a unification has been explored before in the context of the neural theorem prover (NTP) (Rocktäschel & Riedel, 2017); however, NTPs have difficulty scaling to real problem sizes and do not inherently have the ability to operate over natural language. We show how a transformer can effectively translate a natural language statement of a problem into a set of soft rules to be queried through an NTP. We also describe straightforward changes to the NTP that improve its learning dynamics and allow it to handle nontrivial rulesets efficiently.

Our system, shown in Figure 1, consists of several steps. First, a pre-trained transformer encodes a set of text rules, and an attentive rule extractor projects the transformer's encodings into soft rule representations. These rules are fed into a backward-chaining search performing soft unification. This algorithm extends standard NTP inference with dynamic pruning and parallel step processing.

We evaluate our architecture on the SimpleLogic dataset from Zhang et al. (2023). On this dataset, vanilla transformers learn spurious correlations and achieve perfect "in-distribution" accuracy, but fail to generalize at higher proof depths to problems with the same logic sampled from a different process. Our results show that our approach is able to generalize across this distribution gap with no major performance loss, supervised either end-to-end or at the rule level. We also demonstrate the ability to use our architecture's end-to-end differentiability to adapt a model trained on templated rules to more natural text with only example-level labels.

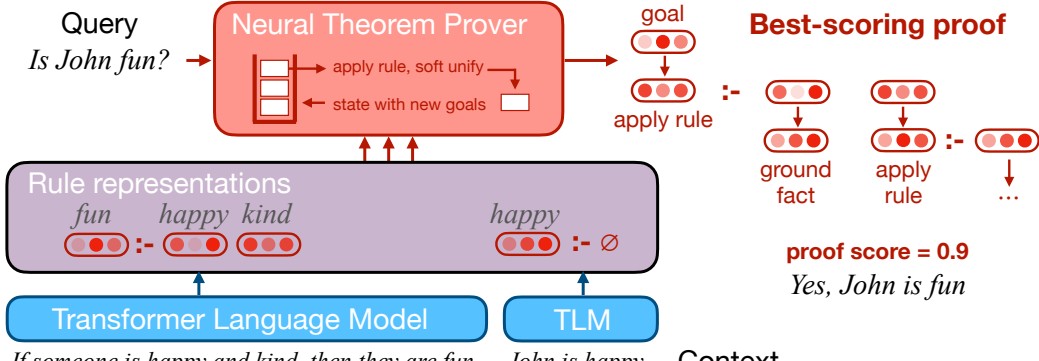

Figure 1: Overview of the PSALM architecture. A Transformer produces an encoding of rules expressed in natural language, which are fed to a neural theorem prover to search over proofs.

Our contributions (1) enable soft proof search with hundreds of rules at higher depths than previously feasible, (2) demonstrate how to differentiably parameterize proof search with transformers, and (3) show that improving architectural inductive bias allows structurally generalizable reasoning to be learned end-to-end.

## 2 BACKGROUND

Zhang et al. (2023) observe that TLMs trained on deduction problems learn incidental statistical features related to the number of rules and facts, and that the ability of these models to predict provability of goals collapses when they are tested on instances of the same problem drawn from a new distribution where these trends no longer hold. This means that the decision functions they acquire conflate aspects of the problem that we hold independent.

We set out to solve this issue architecturally: we would like to modify the computational structure of the TLM to make it easier (or even possible) for the system to learn a deductive decision function that behaves correctly across distributions, while maintaining the softness and learnability of predictions.

The basic hierarchical structure of automated deduction algorithms is backward chaining (Russell & Norvig, 2020, pg. 230), which attempts to find a proof for a goal expression as follows: consequents of the available rules are matched against the current goal, and the antecedents of any matching rules are then introduced as additional subgoals, until all open goals are discharged and search succeeds or all options are exhausted and search fails. This procedure is carried out almost verbatim by the Prolog automated deduction system and logical programming language (Van Emden & Kowalski, 1976; Kowalski, 2014), and forms the backbone of many more advanced deduction systems.

### 2.1 THE NEURAL THEOREM PROVER

Rocktäschel & Riedel (2017) introduce the neural theorem prover (NTP), a differentiable module with an inference procedure analogous to the backward chaining proof search strategy used by Prolog. Prolog rules are Horn clauses $h :- b_1 \wedge b_2 \wedge .. \wedge b_n$. The :- connective is equivalent to $\leftarrow$, a leftward implication: if all the *body* terms $b_i$ are true, then the *head* $h$ is true. A rule can have zero body terms, in which case it is simply a fact: an assertion that its head term is true.

The core inference rule in Prolog is unification: an open goal can be discharged by *applying a rule* if the goal syntactically matches the head of the rule, after which that rule's body terms are introduced as subgoals. Unification in symbolic systems is a discrete operation: either it succeeds, returning a variable assignment under which the two terms are equal, or it fails. The NTP relaxes this discreteness by representing terms as vectors in $\mathbb{R}^d$, making a rule a collection of vectors:

$$\mathbf{h} :- \mathbf{b}_1 .. \mathbf{b}_n \tag{1}$$

The NTP replaces symbolic unification's exact syntactic comparisons with inner products. If the rule above were applied to a goal term vector $\mathbf{g}$, the unification would result in a score of $\mathbf{h} \cdot \mathbf{g}$. We would then have $\mathbf{b}_1, \ldots, \mathbf{b}_n$ as new subgoals to prove by applying additional rules.

In Prolog, a proof is successful if all the unifications involved are successful. In the NTP, a proof can be considered "successful" as long as it is well-formed in that there are no open subgoals. Instead of complete proofs having a binary notion of success or failure, a proof comes with real-valued score, defined to be the minimum over its unification scores, intuitively its weakest link. Let $\mathbf{h}_i$ be the head term vector of the $i$-th rule. Let $\mathbf{b}_{i,k}$ be the $k$-th body term vector of the $i$-th rule. A proof $P$ consists of an applied rule $r(P)$ and subproofs $S_k(P)$, one for each body term of the applied rule. The proof score of the NTP proof $P$ with goal $\mathbf{g}$ is then:

$$\text{score}_{\text{pr}}(P, \mathbf{g}) = \min(\mathbf{h}_{r(P)} \cdot \mathbf{g}, \min_k \text{score}_{\text{pr}}(S_k(P), \mathbf{b}_{r(P),k})) \tag{2}$$

The overall score returned by the NTP process is the maximum score over all considered proofs. As there can be an unlimited number of proofs given one or more non-fact rules, search must be truncated; we impose an upper limit on proof depth and the number of visited search states.

Weber et al. (2019) and Minervini et al. (2020a), among others, have sought to adapt the NTP to natural language inputs. However, the systems proposed in prior work require pipelined rule prediction and continue to suffer from exploding computational cost with increased proof depth. Our experiments also corroborate the findings of De Jong & Sha (2019): the hard minimum and maximum in the NTP scoring function prevent effective exploration of the space of rule representations during training. We describe our approaches to mitigate these issues in the following sections.

## 3 METHODS

The PSALM architecture has two main components: a transformer language model, which encodes input text into continuous rule representations, and a search module, which performs inference based on the encoded rules in order to make predictions.

### 3.1 RULE ENCODING

The rule encoder is responsible for predicting rule representations (1) whose term unification scores reflect the semantic compatibility of rule consequents and antecedents: unifying similar terms should result in a high score, and unifying incompatible terms should result in a low score.

Rules may have variable arity on the right hand side. For example, a statement "*Alice is tall*" has no preconditions, but a statement "*If Alice is open-minded and polite, then she is agreeable*" needs two body terms. Rather than modeling this as a hard decision, we do it softly, predicting rules of all arities simultaneously, only some of which will be used. Specifically, the rule extractor takes the hidden state vectors of the transformer as input, and yields a set of candidate rules by predicting term vectors to fill the slots of $M$ different rule templates. We use four rule templates of the form shown in (1), one for each $n \in [0..3]$. The rule extractor predicts one instance of each rule template per input sentence. The rule encoder can accommodate the unused rules (e.g., an arity 2 rule for the sentence "*Alice is tall*") by learning to assign term vectors with universally low unification scores to the head slots of these inactive rules, preventing them from appearing in high-scoring proofs.

**Predicting rules from TLMs** Let $\mathbf{x} = x_1, \ldots, x_n$ be a sequence of tokens. By providing $\mathbf{x}$ to a TLM and extracting the resulting hidden vectors before its output layer, we can obtain a sequence of embeddings $E = \mathbf{e}_1, \ldots, \mathbf{e}_n$. We pass the embedding sequence through a learnable projection to produce a query, key, and value vector $\in \mathbb{R}^d$ at every token for each term slot in the rule templates. Let $M$ be the number of templates, and let $|T_m|$ be the number of term vectors in the $m$-th template:

$$\forall i \in [1..n], j \in [1.. \sum_{m}^{M} |T_m|]. \ \mathbf{q}_{i,j} : \mathbf{k}_{i,j} : \mathbf{v}_{i,j} = W_j^\top \mathbf{e}_i \tag{3}$$

Let $K_j$ and $V_j$ be the matrices formed by stacking each $\mathbf{k}_{i,j}$ and $\mathbf{v}_{i,j}$ across the sequence. We then apply standard scaled dot product attention to yield a single term vector for each term slot:

$$\mathbf{t}_{i,j} = \text{attn}(\mathbf{q}_{i,j}, K_j, V_j) \tag{4}$$

This can be construed as a multi-headed attention where each term slot is a head. Once we have term vectors $\mathbf{t}_{i,j}$ for each term slot $j$, we can extract rule representations at token $i$ by iterating over rule

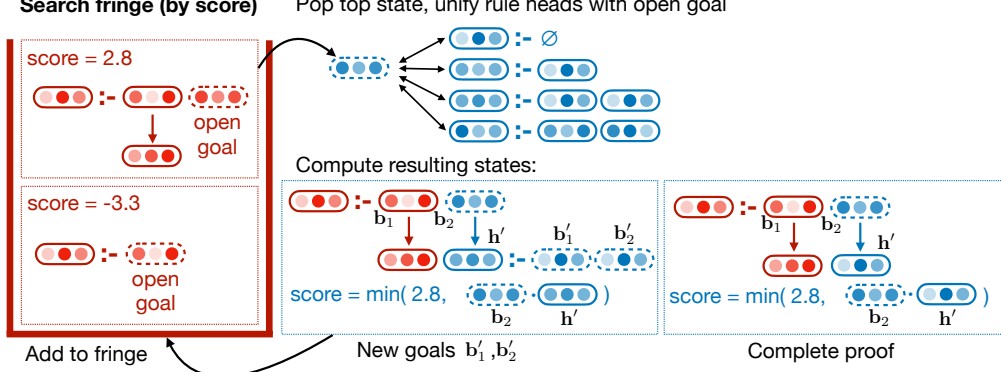

Figure 2: A snapshot of the search procedure, depicting a partial proof undergoing a rule application, as well as the fringe containing all active partial proofs sorted by their lowest soft unification score.

templates and term slots in parallel with predicted term vector indices $j$ and assigning successive $\mathbf{t}_{i,j}$ to each head/body slot. For example, if we had two templates $\mathbf{h}_1 :-$ and $\mathbf{h}_2 :- \mathbf{b}_{2,1}, \mathbf{b}_{2,2}$, we would predict four term vectors $\mathbf{t}_{i,j}$ for $j \in [1..4]$; we would assign $\mathbf{t}_{i,1}$ to $\mathbf{h}_1$, $\mathbf{t}_{i,2}$ to $\mathbf{h}_2$, $\mathbf{t}_{i,3}$ to $\mathbf{b}_{2,1}$, and $\mathbf{t}_{i,4}$ to $\mathbf{b}_{2,2}$, producing the instantiated rules $\mathbf{t}_{i,1} :-$ and $\mathbf{t}_{i,2} :- \mathbf{t}_{i,3}, \mathbf{t}_{i,4}$.

**Split Rule Encoding** We assume for the experiments in this work that each sentence corresponds to a rule, so we encode embedding sequences $E$ and perform the attention operation in (4) separately for each input sentence, and only compute $\mathbf{t}$ vectors (and thus rules) for the final token in each sentence. Encoding rules independently prevents the TLM from "shortcutting" the NTP; put another way, split rule encoding helps us restrict the hypothesis space during learning to generalizable solutions that use the search module to synthesize premise information.

## 3.2 SEARCH

Our architecture relies on the soft proof search procedure to perform reasoning. This procedure, which we describe in Algorithm 1 and show more abstractly in Figure 2, is derived from the NTP algorithm of Rocktäschel & Riedel (2017) described in §2.1. Two major changes are required to make this algorithm practical at our scale: pruning partial proofs, and parallelizing unification.

**Pruning** As we conduct search, early complete proofs provide a useful lower bound on proof scores. We can immediately abandon a partial proof as soon as a rule is applied whose unification score is lower than the current best-scoring proof, as the score of a proof is the minimum across its unification scores, and we only need Algorithm 1 to return the proof with the highest possible score.

As originally described, the NTP did not feature pruning (Rocktäschel & Riedel, 2017). Inefficiency posed a problem for its applicability to real problems, motivating subsequent work to restrict the number of instantiated rules (Minervini et al., 2020b, i.a.). However, restricting the number of rules doesn't solve the underlying issue of being unable to abandon partial proofs. Dynamic pruning allows us to extend the depth of search dramatically beyond what would otherwise be possible, even while maintaining large rule sets.

**Parallelism** The naïve recursive implementation of backtracking search is ill-suited to modern compute hardware, as it visits search states in series. In order to take advantage of GPU parallelism, we design Alg. 1 so that the unification operation can be performed for multiple search states at the same time, as opposed to interleaved with each state's visitation logic. This allows term comparisons, which in our case translate to inner products, to be executed on the GPU as larger vectorized operations without incurring separate dispatch overhead for each term vector.

**Predicting provability** We write $\hat{y} = \sigma(\text{NTP}(\mathbf{g}(x)))$ to denote the final prediction score. We classify examples with a score above $\tau = 0.5$ as provable.

---

**Algorithm 1** Our version of the NTP search routine

---

1: **Inputs:**
2: Rules $\{\mathbf{h}_r \; :\!- \; \mathbf{b}_{r,1}..\mathbf{b}_{r,|B_r|} \mid r \in [1..n]\}$ with $\mathbf{h}_r, \mathbf{b}_{r,i} \in \mathbb{R}^d$
3: Goal $\mathbf{g} \in \mathbb{R}^d$
4: budget $\in \mathbb{N}$
5: maxDepth $\in \mathbb{N}$
6: Unification batch size $b \in \mathbb{N}$
7: **define** states $s \in \mathcal{S}$ to be either the empty state $\emptyset$ or to contain:
8:      Open goals: $\text{goals}(s) = [\mathbf{g}_1..\mathbf{g}_k \in \mathbb{R}^d]$
9:      Score: $\text{score}(s) \in \mathbb{R}$
10:      Best subproof score: $\text{best}(s) \in \mathbb{R}$
11:      Parent state: $\text{parent}(s) \in \mathcal{S}$            ▷ *State whose first open goal this state closes*

12: **let** $\text{depth}(s \in \mathcal{S}) = \begin{cases} \textbf{if } \text{parent}(s) = \emptyset & 0 \\ \textbf{else} & 1 + \text{depth}(\text{parent}(s)) \end{cases}$

13: **let** $\text{lowerBound}(s \in \mathcal{S}) = \begin{cases} \textbf{if } \text{parent}(s) = \emptyset & \text{best}(s) \\ \textbf{else} & \max(\text{best}(s), \text{lowerBound}(\text{parent}(s))) \end{cases}$

14: **function** APPLY(rule $r \in [1..n]$, state $s \in \mathcal{S}$, rule score $u \in \mathbb{R}$)
15:      $o \leftarrow \text{State}(\text{score} = \min(\text{score}(s), u), \text{best} = -\infty)$
16:      **if** $\text{score}(o) < \text{lowerBound}(s) \vee (\text{depth}(s) = \text{maxDepth} \wedge |B_r| > 0)$ **then**
17:          **return** $\emptyset$        ▷ *Prune if score too low or subgoals would break depth limit*
18:      **if** $|B_r| = 0$ **then**             ▷ *Rule is a fact, we can close a subproof*
19:          $c \leftarrow s$
20:          **while** $c \neq \emptyset \wedge |\text{goals}(c)| = 1$ **do**      ▷ *Find ancestor with more than 1 open goal*
21:              $\text{best}(c) \leftarrow \max(\text{best}(c), \text{score}(o))$       ▷ *Update ancestor score bounds*
22:              $c \leftarrow \text{parent}(c)$
23:          **if** $c = \emptyset$ **then**
24:              $\text{goals}(o) \leftarrow []$        ▷ *No ancestors with more than 1 goal, proof is complete*
25:              $\text{parent}(o) \leftarrow \emptyset$
26:          **else**
27:              $\text{goals}(o) \leftarrow [\mathbf{g}_i \mid \mathbf{g}_i \in \text{goals}(c) \wedge i > 1]$     ▷ *Subproof is complete, close 1 goal*
28:              $\text{parent}(o) \leftarrow \text{parent}(c)$
29:      **else**            ▷ *Rule has body terms, so we introduce them as subgoals*
30:          $\text{goals}(o) \leftarrow [\mathbf{b}_{r,i} \mid 1 \leq i \leq |B_r|]$
31:          $\text{parent}(o) \leftarrow s$
32:      **return** $o$

33: **procedure** SEARCH(goal $\mathbf{g} \in \mathbb{R}^d$)
34:      $s_{\text{init}} \leftarrow \text{State}(\text{goals} = [\mathbf{g}], \text{score} = \infty, \text{best} = -\infty, \text{parent} = \emptyset)$
35:      visits $\leftarrow 0$
36:      fringe $\leftarrow \{s_{\text{init}}\}$
37:      **while** visits $<$ budget **do**
38:          $\text{stateBatch} \leftarrow \underset{s \in \text{fringe}, \, k=b}{\operatorname{argtopk}} \text{score}(s)$
39:          fringe $\leftarrow$ fringe $\setminus$ stateBatch
40:          visits $\leftarrow$ visits $+ |\text{stateBatch}|$
41:          $\text{stepScores}[r, s] \leftarrow \mathbf{h}_r \cdot \text{goals}(s)_1 \;\; \forall r \in [1..n], s \in \text{stateBatch}$ ▷ *Batched dot product*
42:          **for** $(r, s) \in \text{stepScores}$ **do**
43:              $s' \leftarrow \text{APPLY}(r, s, \text{stepScores}[r, s])$
44:              **if** $s' = \emptyset$ **then**
45:                  **continue**
46:              **else if** $|\text{goals}(s')| = 0$ **then**
47:                  **yield** $\text{score}(s')$
48:              **else**
49:                  fringe $\leftarrow$ fringe $\cup \{s'\}$

50: **let** $\text{NTP}(\mathbf{g} \in \mathbb{R}^d) = \underset{v \in \text{SEARCH}(\mathbf{g})}{\max} v$

---

# 4 LEARNING

The PSALM NTP module is fully differentiable: unification scores and proof scores have well-defined (sub)derivatives with respect to the TLM's parameters. This offers us several points at which we can potentially apply supervision during training in order to achieve the kind of reasoning behavior we want out of the system. We consider four different objectives at three levels of granularity: examples, proofs, and rules.

## 4.1 END-TO-END

The simplest way to train a PSALM is to leave all intermediate structure latent and optimize end-to-end for proving correct statements and not proving incorrect statements. We do this by applying a binary cross entropy loss, $\mathcal{L}_{\text{E2E}}$, on the predicted proof score (see Algorithm 1) against the example label $y$ (provable or not): $\mathcal{L}_{\text{E2E}}(x) = y \log \sigma(\hat{y}) + (1 - y) \log(1 - \sigma(\hat{y}))$.

We will show in Section 6 that this objective is not sufficient to learn the right latent structure. As noted by De Jong & Sha (2019), this is due to the sparsity of the gradient flow in the vanilla NTP definition: at each training step, only a single proof step's unification score actually receives non-zero gradient in the backward pass. They propose pooling proof scores across multiple alternate proofs; we take this insight a step further. We relax the hard maximum over proof scores to a smooth maximum over the top $k$ highest scoring proofs, but we also relax the hard minimum over step scores to a smooth minimum and add a small amount of Gaussian noise to the unification scores. The corresponding modifications to Algorithm 1 are described in Appendix A.2; we refer to the end-to-end loss using these modifications as $\mathcal{L}_{\text{E2ER}}$.

## 4.2 PROOF DEMONSTRATIONS

The score $\text{NTP}(\mathbf{g}(x))$ produced by the NTP algorithm is not normalized in any way: the stepwise scores are not locally-normalized probability distributions, nor do we view the NTP as placing a globally-normalized distribution over proofs. However, at training time we can still choose to treat the proof process as a generative one and optimize to maximize the likelihood of a collection of gold proof demonstrations. We locally normalize the rule application scores with $\text{softmax}$, yielding a distribution over rule applications at each step, where $\mathbf{g}$ is the current goal term vector. We can then define the probability of a proof as the product of its rule application probabilities:

$$
p_{\text{rule}}(i \mid \mathbf{g}) = \frac{e^{\mathbf{h}_i \cdot \mathbf{g}}}{\sum_{j=1}^{n} e^{\mathbf{h}_j \cdot \mathbf{g}}}
$$

$$
p_{\text{proof}}(P \mid \mathbf{g}) = p_{\text{rule}}\left(r(P) \mid \mathbf{g}\right) \prod_k p_{\text{proof}}\left(S_k(P) \mid \mathbf{b}_{r(P),k}\right)
$$

(5)

Given a set of gold symbolic rules corresponding to the sentences in an example $x$, we can then construct a reference proof $P_{\text{ref}}(x)$ by using symbolic inference, then mapping symbolic rules to the soft rules extracted from their respective sentences with the same number of body terms. The root goal $\mathbf{g}(x)$ is the goal term vector provided by the rule extractor. The demonstration objective $\mathcal{L}_{\text{demo}}(x)$ is the negative log-likelihood loss over these reference proofs $P_{\text{ref}}(x)$:

$$
\mathcal{L}_{\text{demo}}(x) = -\log p_{\text{proof}}\left(P_{\text{ref}}(x) \mid \mathbf{g}(x)\right)
$$

(6)

## 4.3 RULE REPRESENTATIONS

If we have reference symbolic rules, we can supervise rule representations directly. Our rule representation loss $\mathcal{L}_{\text{rule}}(x)$ computes the symbolic unification results between all reference rule head and body terms, and encourages the soft unification scores between rule term vectors to align with those of their discrete counterparts. For instance, in Figure 1, we want to encourage the first term vector (*happy*) to have a similar representation to the *happy* term vector in the second rule, even though the NTP treats both of these as latent vectors with distinct parameters.

We assume a setting where we have $N$ sentences as in Figure 1, where each sentence maps to exactly one symbolic rule. Let $|B_i|$ be the number of body terms in the $i$-th rule. We construct the target matrix $\mathcal{T}$ with the results of symbolic unification of all reference head terms $h_i$ against all body terms $b_{i,j}$ and the goal $g$, with cells equal to 1 where unification succeeds and 0 otherwise:

$$\mathcal{T}(x) = \begin{bmatrix} \text{unify}(h_1, b_{1,1}) & \dots & \text{unify}(h_N, b_{1,1}) \\ \vdots & \ddots & \vdots \\ \text{unify}(h_1, b_{N,|B_N|}) & \dots & \text{unify}(h_N, b_{N,|B_N|}) \\ \text{unify}(h_1, g) & \dots & \text{unify}(h_N, g) \end{bmatrix} \quad (7)$$

Note that $\mathcal{T}$ is a statement about *symbolic* unification and does not yet relate to the soft rules. Let $M$ be the number of soft rule templates; in this setup we have $NM$ total soft rules, not all of which should be active. Let $\phi : [1 .. N] \to [1 .. NM]$ be a mapping from symbolic rule indices to soft rule indices. We define $\phi[i]$ to be the index of the soft rule from the $i$-th sentence with the same number of body terms as the $i$-th symbolic reference rule, i.e. the one soft rule that should be active among those predicted from that location. We construct a soft unification matrix $\mathcal{U}$ over active rule term vectors and the predicted goal vector to align with $\mathcal{T}$:

$$\mathcal{U} = \begin{bmatrix} \mathbf{h}_{\phi[1]} \cdot \mathbf{b}_{\phi[1],1} & \dots & \mathbf{h}_{\phi[N]} \cdot \mathbf{b}_{\phi[1],1} \\ \vdots & \ddots & \vdots \\ \mathbf{h}_{\phi[1]} \cdot \mathbf{b}_{\phi[N],|B_N|} & \dots & \mathbf{h}_{\phi[N]} \cdot \mathbf{b}_{\phi[N],|B_N|} \\ \mathbf{h}_{\phi[1]} \cdot \mathbf{g} & \dots & \mathbf{h}_{\phi[N]} \cdot \mathbf{g} \end{bmatrix} \quad (8)$$

Broadly speaking we want to encourage high values for entries of $\mathcal{U}$ corresponding to valid symbolic unifications in $\mathcal{T}$. We can represent this in an objective as:

$$\mathcal{L}_{\text{rule}}(x) = \frac{1}{|\mathcal{U}|} \sum_{i,j} \mathcal{T}_{i,j} \log \sigma(\mathcal{U}_{i,j}) + (1 - \mathcal{T}_{i,j}) \log(1 - \sigma(\mathcal{U}_{i,j})) \quad (9)$$

This objective only accounts for the soft rules with the right shape; i.e., a rule with two body terms for the first sentence in Figure 1. In order to learn to downweight inactive rules, we concatenate additional columns onto $\mathcal{U}$ for inactive rule head unifications, and add corresponding zeroes to $\mathcal{T}$. As the label distribution in $\mathcal{T}$ can be highly unbalanced, we also apply a rebalancing weight equal to the ratio of the number of 0 labels to the number of 1 labels. The augmented forms of the $\mathcal{U}$ and $\mathcal{T}$ matrices, along with the rebalanced objective, are given in appendix §A.1.

## 5 EXPERIMENTS

We evaluate PSALMs trained with each of the objectives described in §4, a vanilla TLM trained to predict provable/not provable labels directly, as well as the proprietary OpenAI GPT-4o (0-shot, temperature 0) and o1-preview systems. As $\mathcal{L}_{\text{demo}}$ does not produce appropriate $\hat{y}$ scores on its own, we also evaluate a combination of $\mathcal{L}_{\text{demo}}$ and $\mathcal{L}_{\text{E2E}}$ which undergoes an initial round of training under $\mathcal{L}_{\text{demo}}$ followed by a round of fine-tuning with $\mathcal{L}_{\text{E2E}}$ added to its objective.

We conduct an additional comparison in which we first train a PSALM on templated data with $\mathcal{L}_{\text{rule}}$, then fine-tune it on a smaller amount of paraphrased data with $\mathcal{L}_{\text{E2E}}$, comparing it to a vanilla TLM trained with the same data recipe.

All systems we train use DeBERTa v3 Large (He et al., 2023) as the base TLM, with 435M parameters. The PSALM rule extractor adds 4M parameters. We execute PSALM inference with $\text{budget} = 1024$ and unification state batch size $b = 4$[1]. We train models using the Adam optimizer (Kingma & Ba, 2015) with a learning rate of $1e{-}5$ and 1000 steps of linear learning rate warmup followed by linear learning rate decay over 24k total steps with a batch size of 8.

Our primary metric of interest is prediction accuracy (whether $\hat{y} = y$), more specifically accuracy under distribution shift. We also evaluate the soundness of proofs predicted by PSALM systems by translating soft proofs back to discrete ones using the inverse of the discrete→soft rule mapping $\phi$ from §4.3.

---

[1]Note that batched unification also applies all rules in parallel, resulting in a larger effective batch size. On average, unification batches in our training data contain ∼400 term vectors. We experimented with only batching over rules ($b = 1$) as well as larger batch sizes, finding that $b = 4$ yielded the best inference speed.

Table 1: System performance in-distribution (**ID**) on held out rule-priority samples from depths 0-4 and out-of-distribution (**OOD**) on label-priority samples from depths 5-6. The vanilla TLM does not predict proofs and is therefore excluded from soundness comparisons. *The OOD split is imbalanced in the opposite direction from the ID split, and systems can err towards the minority class. †$\mathcal{L}_{\text{demo}}$ does not supervise proof scores with respect to $\tau$, so training with this objective alone is not enough for classification.

| System | ID acc. | ID soundness | OOD acc. | OOD soundness |
|---|---|---|---|---|
| Majority class | 66.6 | – | 76.7 | – |
| Vanilla TLM | 99.6 | – | 64.0 | – |
| GPT-4o | – | – | 79.6 | – |
| o1-preview | – | – | 96.0 | – |
| PSALM-$\mathcal{L}_{\text{E2E}}$ | 76.2 | 3.9 | 23.3* | 0.0 |
| PSALM-$\mathcal{L}_{\text{demo}}$ | 66.9† | 58.1 | 23.3† | 8.6 |
| PSALM-$\mathcal{L}_{\text{demo}}+\mathcal{L}_{\text{E2E}}$ | 82.3 | 64.1 | 28.4* | 26.2 |
| PSALM-$\mathcal{L}_{\text{E2ER}}$ | **100.0** | 100.0 | **99.9** | 98.1 |
| PSALM-$\mathcal{L}_{\text{rule}}$ | **100.0** | 100.0 | 96.7 | 86.3 |

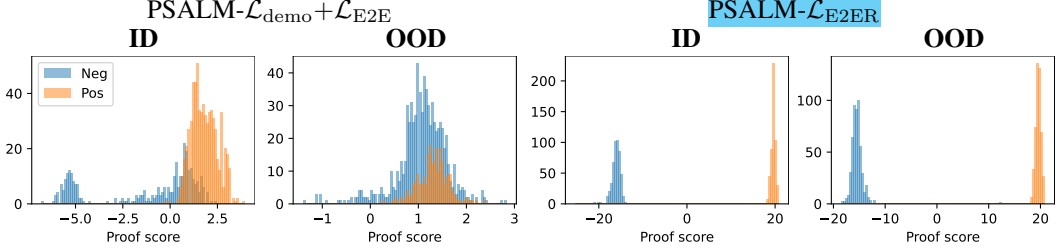

Figure 3: Proof score spreads on each problem distribution for our two end-to-end system variants.

## 5.1 DATA

We train and evaluate systems on the SimpleLogic task of Zhang et al. (2023), a synthetic task where a system is given a set of text rules and facts and must predict whether a query statement holds under the premises. An example of the task format is shown in Figure 7. SimpleLogic examples can be sampled using multiple algorithms. The rule-priority algorithm (RP) first samples rules and facts randomly, then computes the label via forward-chaining deduction. The label-priority algorithm (LP) first samples whether or not particular predicates are true or false, then derives premises that are compatible with this truth table. Samples from each algorithm are formatted identically and follow the same decision rule. However, each algorithm leaves its own statistical traces: for example, the probability of an example's label being positive under the RP algorithm grows as the number of rules increases, but this doesn't hold for the LP algorithm.

We train on 10,000 samples from the RP algorithm sampled to have balanced gold proof depths between 0 and 4. We evaluate on 1,000 held-out samples from this distribution (**ID** in Table 1) as well as 1,000 samples with gold proof depths of 5 and 6 sampled from the LP algorithm (**OOD**). We additionally generate **ID-Para** and **OOD-Para**, sets of 1,000 examples each from the **ID** and **OOD** splits respectively whose premises and queries have been automatically paraphrased with `gpt-3.5-turbo`. An example from **ID-Para** is shown in Figure 8.

## 6 RESULTS

Table 1 shows that a PSALM system with the $\mathcal{L}_{\text{E2ER}}$ objective yields the best out-of-distribution performance of the systems we compare. As shown in Figure 9, when examples get complex enough, even strong pre-trained models can struggle due to autoregressive commitment to the wrong derivation. The strong performance of o1-preview shows that this can be avoided through search, but

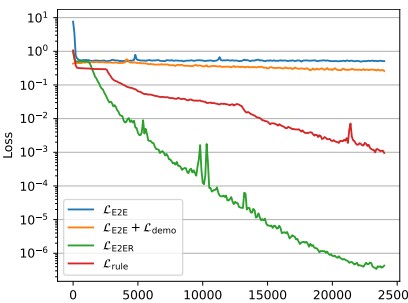

Figure 4: Convergence of each of the objectives we consider.

Table 2: System performance out-of-distribution (**OOD-Para**) on paraphrased label-priority samples from depths 5-6 after end-to-end fine-tuning on 1,000 examples of **ID-Para**, except for PSALM-$\mathcal{L}_{\text{rule}}$ which is not fine-tuned.

| System | OOD-Para acc. |
|---|---|
| Majority class | 61.0 |
| Vanilla TLM | 55.0 |
| PSALM-$\mathcal{L}_{\text{rule}}$ | 50.5 |
| PSALM-$\mathcal{L}_{\text{rule}}+\mathcal{L}_{\text{E2E}}$ | **78.6** |

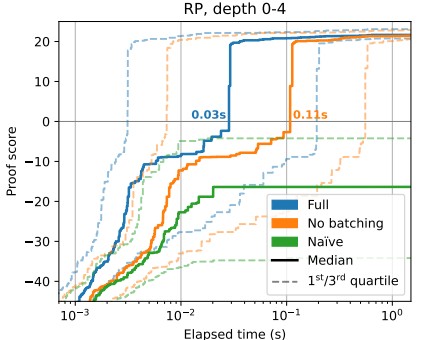
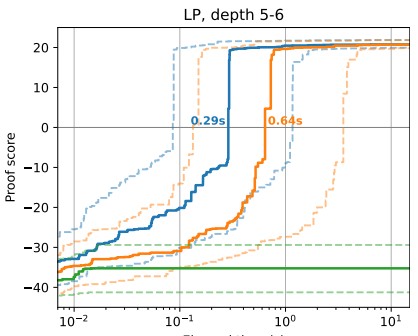

Figure 5: Inference profiles of Algorithm 1 (**Full**) along with two ablations: **No batching** over states and rules (serial unification), and the original NTP algorithm (**Naïve**) without pruning or batching. All variants are profiled over 100 positive (provable) instances and the running maximum over proof scores is recorded for each example.

applying search in token space is expensive; o1-preview takes nearly a minute to complete a single example.

The basic $\mathcal{L}_{\text{E2E}}$ quickly degenerates to predicting a trivial depth-0 proof for all examples, and cannot escape this solution region as shown in Figure 4. $\mathcal{L}_{\text{demo}}$ on its own yields unbalanced scores that do not respect $\tau$ and cannot be used to classify examples as provable or not. When the two are combined, the resulting system is able to avoid both pathological behaviors, but the solution this system converges to is suboptimal compared to the solution found by the relaxed end-to-end objective; while the proof demonstration supervision is able to pull the model out of a poor local minimum at initialization, it also provides a confounding signal preventing the model from converging to optimal label prediction.

Figure 3 shows the shift in scores assigned to provable and unprovable examples by PSALM-$\mathcal{L}_{\text{demo}} + \mathcal{L}_{\text{E2E}}$ and PSALM-$\mathcal{L}_{\text{E2ER}}$ when transferring to the OOD setting. While class balance shifts betwen ID and OOD, no qualitative change in scoring pattern is visible when PSALM-$\mathcal{L}_{\text{E2ER}}$ is applied to a new problem distribution. This supports our hypothesis that the model's inductive bias leads it to a solution that is not dependent on spurious statistical features of its training distribution.

**Generalization to paraphrased data** The setting in Table 1 does not feature the kind of lexical diversity a full TLM would be needed to understand. Table 2 shows results on the paraphrased data. We find that end-to-end label fine-tuning on a small amount of **ID-Para** data is enough to adapt a rule-supervised model to handle paraphrased rules with variable syntax and predicate synonymy. While fine-tuning a vanilla TLM on the same amount of **ID-Para** data does not exceed majority-class on **OOD-Para**, training a PSALM end-to-end on examples from one distribution does yield performance gains out-of-distribution.

**Inference cost**  In Figure 5, we highlight the impact of the changes to the NTP presented in §3.2 and Algorithm 1. Without pruning, the **naïve** algorithm fails to find any positive-scoring proofs within the state budget, making it unusable for problems of this size. Adding dynamic pruning makes it possible to reach successful proofs within 1024 states, with almost all depth 6 proofs reachable in under 4 seconds by the **no batching** ablation. Batching unification then provides a substantial performance boost, bringing most successful depth 6 proofs under 1 second; across all depths, median time to successful proof is at least halved.

On shallower examples from depths 0-4, every successful proof is reachable in under 0.15s, making it practical to perform search during training even with an average of 100 active rules per example and subgoal branching factors up to 3. An additional breakdown of inference speed by depth limit is presented in Figure 6 in the appendix; while the worst-case complexity of our algorithm is still exponential, dynamic pruning prevents this cost from being felt in the vast majority of cases.

# 7 RELATED WORK

A significant line of work has sought to augment LLMs with external solvers. These include calculators (Gao et al., 2023; Chen et al., 2023), logical solvers like Z3 (Ye et al., 2023; Pan et al., 2023), planners like PDDL (Liu et al., 2023), and probabilistic programming languages (Wong et al., 2023). These systems are differentiated from ours mainly by their "hard" solvers. Because they use non-neural tools, training signal can't be passed back to the model from the system's outputs.

The neural theorem prover (Rocktäschel & Riedel, 2017) was originally motivated by this issue, aiming to support backward chaining proof search in a differentiable way. However, the networks described by Rocktäschel & Riedel are exponential in size with respect to the depth of the 'proofs' required. Subsequent work has sought to make this basic idea practical for larger problems by summarizing or filtering active rules based on the current goal (Minervini et al., 2018; 2020a;b; Morris et al., 2022). Weber et al. (2019) in particular adopt a rule score threshold similar to our pruning policy, although their threshold is not updated recursively, limiting the amount of work that can be avoided as depth increases. Our system also features a richer parameterization of rules computed on the fly by a TLM.

Soft proof search is a neural version of a classic discrete algorithm. In this vein, a line of past work has examined data structures like stacks (Grefenstette et al., 2015; Chen et al., 2020), neural Turing machines (Graves et al., 2014), and neural GPUs (Kaiser & Sutskever, 2015). Compared to these architectures, especially the neural Turing machine, our aim is not to learn a very general computation engine, but to buttress one particularly weak capability of transformer LLMs, namely the ability to do deduction and search. This motivation is shared by other recent work fusing transformers with algorithmic neural modules targeted at reasoning (Bounsi et al., 2024).

Differentiable versions of other logical reasoning procedures have been explored, notably probabilistic predicate logic (Manhaeve et al., 2018; Huang et al., 2021) and natural logic (Feng et al., 2020; Shi et al., 2021), applied to tasks like textual entailment Beltagy et al. (2013). This last approach takes logical forms from a separate semantic parser, isolating problem interpretation from execution. Our approach, in contrast, can make "late" decisions about rule viability during search.

# 8 CONCLUSION

In this paper, we present an augmented transformer that can invoke a neurosymbolic proof search module (a neural theorem prover). The transformer instantiates the parameters of the NTP from text inputs, then executes search to find a soft proof of a query, or returns failure if no proof with a high enough score is found. Our experiments analyze several forms of supervision, finding that end-to-end supervision from labels in concert with a relaxed scoring function is sufficient to learn to parameterize latent rules consistently. In order to tackle problems with dozens of rules, we introduce algorithmic improvements to the neural theorem prover that drastically improve its efficiency. Critically, our architecture is able to generalize across problem distributions where standard end-to-end trained transformers fail.

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

# A APPENDIX

## A.1 FULL RULE LOSS

As a reminder, let $M$ be the number of soft rule templates. We have $NM$ total soft rules, $N$ of which should be active. Let $|B_i|$ be the number of body terms in the $i$-th rule. We define the target matrix $\mathcal{T}$ to contain the results of symbolic unification of all reference head terms $h_i$ against all body terms $b_{i,j}$ and the reference goal $g$, where a cell contains 1 if unification succeeds and 0 otherwise. Let $\phi : [1 .. N] \to [1 .. NM]$ be a mapping from symbolic rule indices to soft rule indices. We define $\phi[i]$ to be the index of the soft rule from the $i$-th sentence with the same number of body terms as the $i$-th symbolic reference rule, i.e. the one soft rule that should be active among those predicted from that location.

$$\text{inactive} = \{i \mid 1 \le i \le NM \land i \notin \phi\}$$

$$\mathcal{U}' = \mathcal{U} : \begin{bmatrix} \mathbf{h}_i \cdot \mathbf{b}_{\phi[1],1} \\ \vdots \\ \mathbf{h}_i \cdot \mathbf{b}_{\phi[N],|B_N|} \\ \mathbf{h}_i \cdot \mathbf{g} \end{bmatrix} \forall i \in \text{inactive}$$

$$\mathcal{T}' = \mathcal{T} : \begin{bmatrix} 0 \\ \vdots \\ 0 \end{bmatrix} \forall i \in \text{inactive}$$

$$w = \frac{|\mathcal{T}'| - \sum \mathcal{T}'}{\sum \mathcal{T}'}$$

$$\mathcal{L}_{\text{rule}}(x) = \frac{1}{|\mathcal{U}'|} \sum_{i,j} w \mathcal{T}'_{i,j} \log \sigma(\mathcal{U}'_{i,j}) + (1 - \mathcal{T}'_{i,j}) \log(1 - \sigma(\mathcal{U}'_{i,j})) \tag{10}$$

## A.2 RELAXED NTP DETAILS

We define smoothmax and smoothmin to be weighted sums with softmax weights:

$$\mathrm{smoothmax}(x_i \in X) = \sum_i \frac{x_i e^{x_i}}{\sum_j e^{x_j}}$$

$$\mathrm{smoothmin}(x_i \in X) = -\mathrm{smoothmax}(-X)$$

To relax the NTP algorithm, yielding $\mathrm{NTP_R}$, we make the following changes to Algorithm 1:

Each state $s$ additionally stores $\mathrm{stepScores}(s)$, a list of each step score $u$ involved in the computation of the running $\mathrm{score}(s)$.

Line 10: $k\text{-best}(s) \in \mathbb{R}^{[1,k]}$

Line 13: **let** $\mathrm{lowerBound}(s) = \begin{cases} \textbf{if } \mathrm{parent}(s) = \emptyset & \min(k\text{-best}(s)) \\ \textbf{else} & \max(\min(k\text{-best}(s)), \mathrm{lowerBound}(\mathrm{parent}(s))) \end{cases}$

Line 21: $k\text{-best}(c) \leftarrow \mathrm{top}\text{-}k(k\text{-best}(c) : [\mathrm{score}(o)])$

Line 41: $\mathrm{stepScores}[r, s] \leftarrow \mathbf{h}_r \cdot \mathrm{goals}(s)_1 + \epsilon \sim N(0, \sigma^2) \; \forall \, r \in [1..n], s \in \mathrm{stateBatch}$

We set the noise scale $\sigma = 0.1$, chosen heuristically.

Line 47: **yield** $\mathrm{smoothmin}(\mathrm{stepScores}(s'))$

Line 50: **let** $\mathrm{NTP_R}(\mathbf{g}) = \mathrm{smoothmax}(\mathrm{top}\text{-}k(\textsc{Search}(\mathbf{g})))$

## A.3 ADDITIONAL RUNTIME ANALYSIS

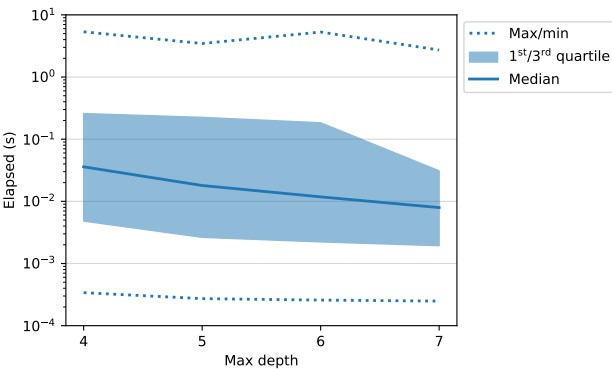

Figure 6: Elapsed time in PSALM search by depth limit (measured over LP samples from depths 0-6). While worst-case time complexity is still exponential, as in the original NTP (reflected in the elapsed time varying across several orders of magnitude by example), dynamic pruning ensures that the bulk of cases are handled efficiently; as search depth increases, tighter score bounds are found to offset the increase in horizon.

## A.4 SIMPLELOGIC SAMPLES

| | |
|---|---|
| Input: | *If someone is foolish and frantic, then they are nervous.* |
| | *If someone is excited, then they are different.* |
| | *Alice is excited.* |
| | *Alice is foolish.* |
| | *Q: Alice is nervous.* |
| | *A:* |
| Label: | *False* |

Figure 7: An example of the SimpleLogic task format.

| | |
|---|---|
| Input: | *Alice has good manners.* |
| | *Someone is versatile if they are both brave and scared.* |
| | *Alice is feeling afraid.* |
| | *If someone is sure of themselves and filled with questions, then they are terrified.* |
| | *If someone possesses excitement, meanness, and courage, then they are considered old-fashioned.* |
| | *Being versatile, hostile, and nervous means that someone is renowned.* |
| | *Someone's reputation for being rude, conservative, and well-known indicates their courage.* |
| | *Alice is adaptable.* |
| | *Q: Alice is feeling nervous.* |
| | *A:* |
| Label: | *True* |

Figure 8: An example from the paraphrased SimpleLogic **ID-Para** split.

| | |
|---|---|
| Input: | *If someone is proud, then they are aggressive.* |
| | *If someone is hurt, bored, and stubborn, then they are long.* |
| | *If someone is stubborn, glamorous, and loving, then they are rude.* |
| | *If someone is vivacious, then they are cruel.* |
| | *If someone is naughty, vivacious, and hurt, then they are sincere.* |
| | *If someone is nervous, polite, and stubborn, then they are dull.* |
| | *If someone is smart, then they are helpful.* |
| | *If someone is powerful and outstanding, then they are wrong.* |
| | *If someone is smart, nervous, and wrong, then they are stubborn.* |
| | *If someone is aggressive and tender, then they are bored.* |
| | *If someone is glamorous, talented, and smart, then they are wrong.* |
| | *If someone is sincere, long, and proud, then they are stubborn.* |
| | *If someone is cruel, then they are condemned.* |
| | *If someone is wrong, then they are scared.* |
| | *If someone is wrong and pleasant, then they are glamorous.* |
| | *If someone is aggressive, then they are horrible.* |
| | *If someone is dull and tender, then they are outstanding.* |
| | *If someone is horrible, nervous, and wrong, then they are polite.* |
| | *If someone is aggressive and precious, then they are sincere.* |
| | *If someone is hurt, then they are loving.* |
| | *If someone is cruel, then they are talented.* |
| | *If someone is powerful and horrible, then they are long.* |
| | *If someone is victorious, then they are frantic.* |
| | *If someone is aggressive, then they are powerful.* |
| | *If someone is talented, then they are rude.* |
| | *If someone is tender, dull, and victorious, then they are frantic.* |
| | *If someone is long, then they are helpful.* |
| | *If someone is frantic, smart, and hurt, then they are cruel.* |
| | *If someone is stubborn, then they are precious.* |
| | *If someone is sincere, then they are glamorous.* |
| | *If someone is hurt, frantic, and scared, then they are powerful.* |
| | *If someone is sincere, then they are smart.* |
| | *If someone is tender, then they are naughty.* |
| | *If someone is vivacious and bored, then they are talented.* |
| | *If someone is loving, then they are cruel.* |
| | *If someone is horrible and dull, then they are powerful.* |
| | *Q: Is Alice powerful?* |
| Label: | *True* |
| GPT-4o: | *To determine if Alice is powerful, we need to follow the logical implications given in the statements:* |
| | *1. Alice is hurt.* |
| | *2. If someone is hurt, then they are loving.* |
| | *3. If someone is loving, then they are cruel.* |
| | *4. If someone is cruel, then they are condemned.* |
| | *5. If someone is cruel, then they are long.* |
| | *6. If someone is long, then they are helpful.* |
| | *7. If someone is long, then they are vivacious.* |
| | *8. If someone is vivacious, then they are cruel.* |
| | *9. If someone is cruel, then they are talented.* |
| | *10. If someone is talented, then they are proud.* |
| | *11. If someone is proud, then they are aggressive.* |
| | *12. If someone is aggressive, then they are horrible.* |
| | *13. If someone is horrible and dull, then they are powerful.* |
| | *From the above deductions, we can see that Alice is hurt, which leads to her being loving, cruel, condemned, long, helpful, vivacious, talented, proud, aggressive, and horrible. However, we do not have information about Alice being dull, which is necessary to conclude that she is powerful based on the statement "If someone is horrible and dull, then they are powerful."* |
| | *Therefore, based on the given information, we cannot definitively conclude that Alice is powerful.* |

The second column of the rule list:

*If someone is smart and glamorous, then they are polite.*
*If someone is helpful, hurt, and polite, then they are proud.*
*If someone is naughty and long, then they are wrong.*
*If someone is long, loving, and precious, then they are cruel.*
*If someone is precious, then they are wrong.*
*If someone is nervous, dull, and proud, then they are bored.*
*If someone is victorious, loving, and long, then they are powerful.*
*If someone is bored, then they are sincere.*
*If someone is precious, then they are glamorous.*
*If someone is horrible, hurt, and scared, then they are outstanding.*
*If someone is talented, dull, and loving, then they are vivacious.*
*If someone is bored, then they are hurt.*
*If someone is talented, condemned, and precious, then they are hurt.*
*If someone is cruel, then they are long.*
*If someone is smart and polite, then they are powerful.*
*If someone is long, then they are vivacious.*
*If someone is aggressive, rude, and wrong, then they are cruel.*
*If someone is smart, then they are stubborn.*
*If someone is cruel and smart, then they are naughty.*
*If someone is scared, then they are helpful.*
*If someone is naughty and outstanding, then they are stubborn.*
*If someone is scared, then they are glamorous.*
*If someone is wrong, then they are victorious.*
*If someone is smart and horrible, then they are hurt.*
*If someone is long, pleasant, and wrong, then they are glamorous.*
*If someone is vivacious and helpful, then they are pleasant.*
*If someone is polite, then they are dull.*
*If someone is talented, then they are proud.*
*If someone is dull and scared, then they are naughty.*
*If someone is stubborn, then they are nervous.*
*If someone is polite, then they are tender.*
*If someone is hurt, then they are rude.*
*If someone is scared and sincere, then they are talented.*
*If someone is aggressive, outstanding, and wrong, then they are bored.*
*If someone is vivacious and sincere, then they are stubborn.*
*Alice is hurt.*

Figure 9: A depth-6 SimpleLogic example and GPT-4o's incorrect prediction. In the final two steps, highlighted in orange, the model commits to a premise that doesn't hold, instead of directly using the premise "*If someone is aggressive, then they are powerful.*" This greedy overcommitment is a shortcoming of systems that do not take advantage of search. In contrast, both PSALM and o1-preview predict the correct label. PSALM takes 0.6 seconds; o1-preview uses 4,307 private inference tokens and takes 46.1 seconds.

## A.5 ADDITIONAL RELATED WORK

There is a close conceptual connection between the differentiable proof structures produced by NTP search and the network structures predicted by neural module networks (Andreas et al., 2016; Gupta et al., 2020). In a neural module network, differentiable components with specialized inductive biases are composed hierarchically on an example-by-example basis. Each module is treated as a function, and the layout of the network is predicted by a semantic parser conditioned on the problem to be solved. Our approach shares the core idea of generalization through component reuse: if a complex problem is made up of simple subproblems, we can structure a model's computational abilities so that the model is able to solve complex instances of a problem class by inferring how to decompose them and applying a set of learned solutions to the elementary subproblems. Strategies like this offer not only better interpretability, but also better data efficiency, as non-compositional models of compositional problems require the training set to capture a much larger range of combinations of properties or steps.

Other work has investigated forward-chaining (Tafjord et al., 2021; Gontier et al., 2020) and backward-chaining search in natural language (Sprague et al., 2022; Tafjord et al., 2022) and in conversational reasoning (Arabshahi et al., 2021). Generating forward-chaining proofs autoregressively is challenging, as models must predict which ground facts to introduce from the bottom up; doing this accurately requires inferring the entire proof tree before it can be emitted. Regardless of expansion order, any form of search over generated strings is challenging, as it can quickly run off the rails due to cascading errors; in contrast, our rule representations don't require decoding to strings and thus allow much more efficient and predictable inference. Approaches like maieutic prompting (Jung et al., 2022) limit divergence by only unrolling one or two steps of reasoning. Weir et al. (2023) achieve better control by specializing backward chaining in natural language to a particular domain with tailored templates, at the cost of domain flexibility.

Yang & Deng (2023) set out in a less-traveled direction, investigating rule learning for reasoning over text *without* gradient descent; their basic inference operation is based on string substitution, making use of *reverse unification* to learn more abstract rules from concrete ones.

