# OpenReview forum: "Proof Search Augmented Language Models"
_ICLR.cc/2025/Conference — Submitted to ICLR 2025_

### Official Review · Reviewer_Td5p · 2024-10-19

**Soundness:** 2
**Presentation:** 2
**Contribution:** 2
**Rating:** 5
**Confidence:** 4

**Summary:**

Summary:

This paper applies an encoder-only model to represent the rule and statement in the proof. Then, the Neural Theorem Prover (NTP) utilizes the representation to perform a backward-chaning search of proofs and sort them.

Contributions:

This paper proposes a new method to improve the model reasoning abilities by semantics proof search. The experimental results on the SimpleLogic task are nice.

**Strengths:**

- This paper proposes a new method to improve the model reasoning abilities by semantics proof search. The experimental results on the SimpleLogic task are nice.

**Weaknesses:**

1. The experiment section is insufficient. First, only one dataset named SimpleLogic is used in the experiment. Second, the author only uses one model in the experiment. Last but not least, there are some work aimed to improve the proof reasoning abilities. However, I have not seen the gap between those work and the method proposed by the authors.

2. The terminology in this article is not to the standard. For instance, we typically refer "transformer" to encoder-decoder architecture models instead of encoder-only models.

**Questions:**

1. Why are language models such as Llama not used in the experiments?
2. If language models are used, do we still need the complex procedure to make it work?

---

> ### Comment · Reviewer_1xBQ · 2024-11-25
>
> I have read all the reviews so far. Since the authors have not addressed any of the concerns or answered any questions, I have lowered my rating slightly to reflect that.

---

> > ### Author Response · Authors · 2024-11-25
> >
> > Sorry for the radio silence! We have read and appreciate all the reviewers' comments and questions -- we're currently working to revise the paper and we will be posting a full response to all of these points once the updated PDF is in place.

---

> ### Author Response · Authors · 2024-11-28
> **Response to Reviewer Td5p**
>
> Thanks for your patience - we've now updated the PDF and posted a general response describing our revisions!
>
> **W1:** See our general response on evaluation scope.
>
> **W2:** While the ‘transformer’ was originally introduced as an encoder-decoder architecture, in contemporary practice it is frequently used interchangeably for encoder-only, decoder-only, or encoder-decoder variants – see e.g. the RoBERTa paper (Liu et al, 2019) p2: “BERT uses the now ubiquitous transformer architecture [...]”
>
> **Q1 and Q2:** See our general response on evaluation scope and our new LLM comparison.

---

### Official Review · Reviewer_1xBQ · 2024-11-03

**Soundness:** 3
**Presentation:** 3
**Contribution:** 3
**Rating:** 5
**Confidence:** 4

**Summary:**

The paper proposes a PSALM model that combines a transformer-based language model with a proof search system. Compared to the NTP introduced by Rocktaschel & Riedel in 2017, it uses pruning and parallel execution to improve scaling and throughputs. Using a new loss term, the system provides significant improvements over a vanilla model for (limited) out-of-distrbution generalization.

**Strengths:**

The paper is well written and the key components are motivated and explained well. The experimental results seem to be systematic and convincing.

**Weaknesses:**

The novelty introduced in the paper seems limited. Pruning is well known; while it improves scaling, in the worst case, the complexity remains the same. Also the improvements over the vanilla model do not reflect the improvements over the state of the art transformer models.

The experimental results are not fully explained. For example, the row of PSALM L_{rule} in Table 1is not explained and no comments are provided in the paper even though it is a key result.

**Questions:**

- Could you explain the reasons why loss L_{rule} is minimized much smaller than the other two? It seems to me it shows that many entries in matrix T are zeroes.

- As a follow-up question, since the loss terms can be minimized to a different scale, should different weights improve the performance when combining them such as the results in Tables 1 and 2?

- Section 3.1 states that "Encoding rules independently prevents the TLM from “shortcutting” the NTP," could you provide corresponding empirical evidence and quantify the impact?

- Figure 3 shows the range of proof score is different from the case on the left and the case on the right. Could you explain how the larger range for L_{rule} affect its generalization and separation between positive and negative ones?

- Could you provide the key reasons underlying the big differences between the last two rows of Table 1 in terms of OOD accuracy and OOD soundness?

- I assume the vanilla TLM model is the DeBerta model with 435M parameters. However, there are manyother larger LLMs developed. Could you provide a baseline or baselines using state of the art LLMs?

- Could you explain how the proposed method overcomes the exploding computational cost compared to other methods? As far as I could understand, the pruning and parallelization do not change the nature of exponential growth.

- Could you comment on the complexity of the proposed system with respect to the depth? In addition to depths 5-6, could you provide results for depths 7, 10, and 20?

---

> ### Author Response · Authors · 2024-11-28
> **Response to Reviewer 1xBQ**
>
> Thanks for your careful consideration and in-depth questions!
>
> **W1:** See our new results in the general response; we compare to GPT-4o and o1-preview and have adopted new NTP training techniques which allow PSALMs to learn SimpleLogic robustly from only end-to-end supervision.
>
> **W2:** We comment on these results in the first paragraph of Section 6.
>
> **Q1 - Could you explain the reasons why loss $L_{rule}$ is minimized much smaller than the other two?:** There is inherent uncertainty in $L_{demo}$, as multiple rules may be valid choices to discharge a particular subgoal; the model has no way of knowing which will occur in the gold demonstration, which leads to a nonzero loss floor for $L_{demo}$. The basic version of $L_{E2E}$ does not converge as it gets stuck in a bad local minimum at initialization. However, we have now fixed that with a modification to the method, as described in our general response. You are right that many entries in the target matrix are zeroes, however, the full $L_{rule}$ rebalances the labels in the objective so that the model does not resort to trivially predicting 0 for every score; see Appendix A.1.
>
> **Q2 - Should different weights improve the performance when combining losses?:** Yes, these losses do have different scales, and reweighting them when combining them is potentially beneficial; however, we did not pursue a detailed hyperparameter search over loss weights in favor of attempting to mitigate the issues with L_E2E alone.
>
> **Q3 - Could you quantify the impact of independent rule encoding?:** In our preliminary experiments, encoding rules jointly allowed the model to learn trivial rule systems consisting of only a single fact, whose unification score with the query encoded the predicted score directly resulting in no actual proof search occurring. This does not happen with $L_{rule}$, as adjacent rules are independent so no shortcuts are learned regardless of their availability; we kept rule encoding independent in the $L_{rule}$ configuration for simplicity.
>
> **Q4 - Could you explain how the larger range for $L_{rule}$ affect its generalization and separation between positive and negative ones?:** The wider score separation for $L_{rule}$ is a reflection of the fact that the model is better able to minimize $L_{rule}$, resulting in more confident unification score predictions. We have updated Figure 3 to show the score separations learned by the new $L_{E2ER}$ configuration, which learns an even cleaner separation.
>
> **Q5 - Could you provide the key reasons underlying the big differences between the last two rows of Table 1 in terms of OOD accuracy and OOD soundness?:** The OOD split has two major differences with the ID split: gold proofs are deeper, and the label imbalance is different - there are now more negatives than positives. Including the original version of $L_{E2E}$ in the loss mixture in the second-to-last row of Table 1 allows the model to learn reasonable score separations for the ID split, but this solution still has many incorrect unification scores (as evidenced by the poor ID soundness). Transferring to the OOD split increases the rate of incidence of these false-negative/false-positive steps per example (as examples are deeper).
>
> **Q6 - Could you provide a baseline or baselines using state of the art LLMs?:** See our general response on our new LLM comparisons. For PSALM itself, larger LLMs are not needed to address the settings in this paper, but will generally be needed in the future as the architecture is scaled to higher complexity of text and more domains.
>
> **Q7 - Could you explain how the proposed method overcomes the exploding computational cost compared to other methods?:** We have not undertaken a theoretical analysis of the impact of pruning or parallelization, but empirically they both confer a major increase in the maximum proof depths we are able to reach in a reasonable amount of time. Pruning in particular is essential, as completing one proof allows the elimination of the bulk of the fringe. In principle, the algorithm is still exponential, as it is possible to construct rule sets for which search continues to find higher-scoring partial proofs very deep in the tree, but in practice examples where this occurs are rare. While pruning is a well-known technique, prior work describing its application in NTPs either only uses it for fact unification (i.e. nearest-neighbor, Minervini et al. 2021), or uses a static threshold, which doesn’t result in as powerful of an effect.
>
> **Q8 - Could you comment on the complexity of the proposed system with respect to the depth?:** See our response to Reviewer xqTT; we have included a breakdown of time by depth in Figure 6 in the appendix. The SimpleLogic depth analysis algorithm (used to generate and bucket examples) is also exponential in complexity by depth - quantifying example depths higher than 6 or 7 becomes impractically slow, so we have limited our analysis to depth 6.

---

### Official Review · Reviewer_xqTT · 2024-11-04

**Soundness:** 2
**Presentation:** 3
**Contribution:** 3
**Rating:** 6
**Confidence:** 4

**Summary:**

The authors introduce Proof Search Augmented Language Models (PSALMs), a differentiable proof search module combined with a transformer. The authors propose an efficient hardware-aware method for proof search (at further depths than prior works) and pruning and performing ablations to identify the strengths of granular rule supervision.

**Strengths:**

- The paper presents multiple training objectives and studies which impact performance.
- An efficient and hardware-aware algorithm has been proposed for proof search.
- The authors show evaluations of the SimpleLogic dataset and generalization capabilities.

**Weaknesses:**

- There are limited empirical evaluations other than SimpleLogic.
- More investigation could be conducted into the scaling of the proof search.

**Questions:**

- How does the approach work on tasks other than SimpleLogic?

---

> ### Author Response · Authors · 2024-11-28
> **Response to Reviewer xqTT**
>
> **W1/Q2**: See our general response on evaluation scope.
>
> **W2**: Our new results include additional analysis of median elapsed time for varying search depths, showing that dynamic pruning prevents search costs from exploding.

---

> > ### Comment · Reviewer_xqTT · 2024-11-30
> >
> > Thank you for your responses. My score will remain unchanged.

---

### Official Review · Reviewer_Hfn3 · 2024-11-04

**Soundness:** 3
**Presentation:** 3
**Contribution:** 2
**Rating:** 5
**Confidence:** 3

**Summary:**

The paper focuses on multi-step reasoning tasks that require a model to predict whether a statement is true given unification rules and facts in natural language. The authors first define several rule templates that contain different numbers of unification terms, then train a Transformer and cross-attention module to fill the term slots with entities and their features from a sentence. Lastly, the extracted rule is input to a neural theorem prover to obtain the proof and the truth prediction. The author uses three types of supervision: label supervision, proof supervision, and rule supervision. Experiments show training with rule supervision can obtain 96.7% accuracy on the OOD test split.

**Strengths:**

The proposed method is effective. The experiments show the proposed method trained with rule labels can obtain nearly 100% accuracy on the OOD splits, which largely improves the previous methods.

**Weaknesses:**

- The evaluation dataset and the compared method are too simple. Since the paper claims current TLMs have limits in reasoning, it should compare with recent SOTA LLMs and show their incapability in a complicated benchmark.

- The comparison is unfair. The proposed method with proof and label supervision achieves inferior performance to baseline TLM and only obtains nearly 100% accuracy with rule-level supervision. A more appropriate baseline should also have these labels. For example, an LLM trained to generate text in "fun :- happy kind" format and predict the label with CoT.

**Questions:**

- Is there any harder QA benchmark that involves logic or multi-step reasoning suitable to evaluate the proposed method?

- How do current LLMs w/wo CoT perform in the SimpleLogic dataset?

---

> ### Author Response · Authors · 2024-11-28
> **Response to Reviewer Hfn3**
>
> **W1/Q2 (Comparison with LLMs):** See the results of our new comparison with GPT-4o and o1-preview in the general response.
>
> **W2 (Unfair comparison):** See the additional results we describe in our general response. We can now outperform o1 with pure end-to-end training.
>
> **Q1 (Other benchmarks):** See our general response on evaluation scope.

---

> > ### Comment · Reviewer_Hfn3 · 2024-12-03
> >
> > Thanks for the response. The new result of end-to-end training is impressive. However, the CoT-LLM (o1) also achieves 96%, which shows that the problem is not challenging for current methods. I still think a harder benchmark, which current LLMs will fail, is necessary to validate the statement. I will keep my scores unchanged.

---

### Author Response · Authors · 2024-11-28
**General response to reviewers**

Thanks to all the reviewers for their feedback!

**New results:**

We have new results with a further modified NTP that is able to achieve 99% OOD accuracy using only end-to-end supervision. We achieve this by replacing the hard min/max in the NTP score aggregation with a soft relaxation, and injecting small amounts of noise in the unification score function to help escape poor local minima at initialization. This removes the previous need for additional supervision in the form of gold symbolic rule systems. We have revised our contributions (line 71), Table 1, discussion in Section 6, and conclusion to reflect these new results. We describe the new objective in Section 4.1 and Appendix A.2.

We have also included GPT-4o (80%) and o1-preview (96%) in our evaluation to provide a strong pre-trained LLM reference point; see Table 1, discussion at the start of Section 6, and Figure 9 in Appendix A.4.

Content changes in the updated PDF are highlighted in cyan.

**Scope of evaluation:**

Our main goal in this paper is to demonstrate a way to overcome the poor inductive bias of transformer models when trained to decide logical reasoning problems. We adopt SimpleLogic as a testbed because it was previously introduced specifically to highlight this shortcoming in vanilla transformers. Other, more difficult logical reasoning benchmarks exist, e.g. P-FOLIO, PrOntoQA, LogicBench, and MUSR. However, the logical structures backing these datasets are richer: P-FOLIO involves nested disjunction and negation, PrOntoQA involves quantification and degree properties, LogicBench involves quantification, negation, disjunction, and defeasibility, and MUSR involves both complex predicate-argument and temporal/causal structure, including implicit defeasible world knowledge. For PSALMs to succeed beyond propositional Horn clause systems (SimpleLogic), we would need to enrich the syntax of the NTP rule structures significantly. This is not an inherent or insurmountable limitation of the PSALM architecture, but rather a pragmatic limitation of the rule projection layer we use in the current version of the system. Changing this would increase the complexity beyond what we feel is appropriate in scope for a single ICLR paper with a focused contribution.

---

### Meta-Review · Area_Chair_5f26 · 2024-12-23

**Metareview:**

The paper proposes Proof Search Augmented Language Models (PSALMs), which combine a transformer-based language model with a proof search system. The authors use three types of supervision: label, proof, and rule supervision. They conduct experiments on the SimpleLogic dataset to demonstrate the effectiveness of their method in overcoming the poor inductive bias of transformer models in logical reasoning problems. However, 1) the evaluation dataset and compared methods are too simple. the high accuracy of CoT-LLM shows the problem may not be challenging enough, and a harder benchmark is needed. 2) Limited novelty as pruning is well-known, and improvements over the vanilla model do not reflect improvements over SOTA transformer models.  Overall, the authors should verify their proposed method with more reasonable benchmarks with more experimental results as well as comprehensive analysis.

**Additional Comments On Reviewer Discussion:**

Multiple reviewers have pointed out the limited scope of evaluation, both in terms of datasets and compared methods. The lack of comparison with state-of-the-art LLMs comprehensively and the use of a relatively simple dataset remain concerns. The novelty of the approach is also questioned, with pruning being a well-known technique. Although the authors include new results in the rebuttal, the reviewers still are not convinced due to the limited content in the experimental section.

---

### Decision · Program_Chairs · 2025-01-22

Reject